# Study on the Work Hardening and Metamorphic Layer Characteristics of Milling Contour Bevel Gears

**DOI:** 10.3390/ma15227975

**Published:** 2022-11-11

**Authors:** Mingyang Wu, Jianyu Zhang, Chunjie Ma, Yali Zhang, Yaonan Cheng, Shi Wu, Lubin Li

**Affiliations:** 1School of Mechanical Engineering, Harbin University of Science and Technology, Harbin 150080, China; 2School of Mechanical Engineering, Heilongjiang University of Science and Technology, Harbin 150020, China

**Keywords:** contour bevel gears, metamorphic layer, work hardening, finite element simulation, XRD diffraction analysis

## Abstract

High temperature and strain will occur in the cutting area during dry milling of contour bevel gears, which causes plastic deformation of the workpiece, resulting in changes in the physical properties of the machined surface’s metamorphic layer, reducing the quality of the workpiece’s machined surface. Therefore, it is necessary to investigate the properties of the metamorphic layer and the work hardening behavior of the machined surfaces of contour bevel gears. The paper first establishes a single-tooth finite element simulation model for a contour bevel gear and extracts the temperature field, strain field and strain rate at different depths from the machined surface. Then, based on the simulation results, the experiment of milling contour bevel gears is carried out, the microscopic properties of the machined metamorphic layer are studied using XRD diffractometer and ultra-deep field microscopy, and the work hardening behavior of the machined metamorphic layer under different cutting parameters is studied. Finally, the influence of the cutting parameters on the thickness of the metamorphic layer of the machined surface is investigated by scanning electron microscopy. The research results can not only improve the surface quality and machinability of the workpiece, but are also significant for increasing the fatigue strength of the workpiece.

## 1. Introduction

The contour bevel gear is a key part in the field of mechanical transmission that transmits motion or power between intersecting shafts or staggered shafts. 20CrMnTi is a commonly used material for contour bevel gears, which produces high cutting temperature and strain during the cutting process, resulting in serious deformation and work hardening of the surface metal on the machined surface. It will lead to uneven distribution of the metamorphic layer formed on the machined surface, and unfavorable factors such as tissue stress may easily occur during the use of the gears, which may cause the metamorphic layer to fall off locally, affecting the surface accuracy, working size, and transmission efficiency. Therefore, it is necessary to study the metamorphic layer characteristics and the work hardening behavior of the machined surface. 

In regards to the machined surface’s metamorphic layer and work hardening behavior, scholars mainly conducted research based on the formation mechanism of a surface’s metamorphic layer, the influence of process factors on surface microstructure, and the simulation and prediction of microstructure [1]. Wang, L. et al. [2] studied the grinding performance and grinding mechanism of a WD-201 microcrystalline corundum grinding wheel on 20CrMnTi steel gears in addition to studying the effects of different metallographic structures on residual stress and hardness through experiments. Gupta, K. et al. [3] studied the variation law of surface profile deviation and surface roughness with discharge energy parameters through EDM machining of micro spur gears and analyzed the effects of wire EDM on the surface morphology, bearing length parameters, microstructure of pinions and hardness. Chen, T. et al. [4] conducted a high-speed hard cutting test on GCrl5 hardened bearing steel and studied the causes of the white layer at different cutting speeds. Pathak, S. et al. [5] evaluated the surface quality of bevel gears by detecting the surface finish, microstructure, and hardness, and found that the surface quality and microgeometric parameters of bevel gears after hardened honing were better than those without hardened honing. Liu, Z. [6] carried out an ultrasonic rolling test of 18CrNiMo7-6 gear steel and explored the change of the metamorphic layer on the machined surface of the gear steel.

Some scholars have also conducted research based on the processing properties of 20CrMnTi materials and the simulation properties of gear steel. Gao, Y. et al. [7] used X-ray diffraction to detect the residual stress of 20CrMnTi after processing and found that the substrate material of 20CrMnTi was a dual-phase alloy of ferrite and pearlite. Li, X. et al. [8] analyzed 20CrMnTi with XRD, obtained XRD patterns and analyzed the proportion of elements. They studied the hot deformation behavior of 20CrMnTi steel at high temperature through isothermal compression test, the coefficient K related to temperature and strain rate was proposed, and the material constitutive equation was established. Zhao, X. et al. [9] proposed a prediction modification method of the constitutive model, and the accuracy of the modification model was verified by isothermal compression test of 20CrMnTi. Wu, S. et al. [10] used Gleeble-3500 thermal simulation technology to study the thermal simulation of 20CrMnTiH steel. The Arrhenius equation was used to establish the thermoplastic constitutive model and the relationship between flow stress, strain, strain rate and temperature in the forging process of 20CrMnTiH was revealed. Han, X. et al. [11] studied the grain evolution process of gear tooth surface with an optical microscope and studied the evolution of cementite particles and gear tooth microstructure during cold rotary forging. Li, L. et al. [12] studied the transformation law of 20CrMnTi during continuous cooling and found that there were three transformation regions of ferrite-pearlite, bainite and martensite in 20CrMnTi during continuous cooling. When the cooling rate increased, the starting time and duration of ferrite transformation in the steel shortened, resulting in an increase in the hardness of 20CrMnTi steel. Zhang, B. et al. [13] studied the relationship between temperature and deformation resistance of 20CrMnTi steel during deformation in a two-phase region with an isothermal compression experiment, and analyzed the microstructure of the deformation region. Lv, L. et al. [14] studied the microstructure of a closed extruding fine blanking spur gear and found that the metal streamline near the extrusion shear surface is relatively dense, the gear material in the deformation area produced large plastic deformation, and the material mainly flows along the axial direction. Yang, J. et al. [15] established the prediction theory of work hardening degree based on low-speed orthogonal metal cutting. Yin, S. et al. [16] developed the 3D/2D hybrid multi-physical-field model to predict the effect of F-EMS on fluid flow and successfully simulated the solute segregation of the continuously cast billet with final electromagnetic stirring. Guo, J. et al. [17], based on the temperature-diffusion-transformation-stress multi-field coupling theory, established the improved multi-field multi-scale coupling model of the vacuum low-pressure carburizing process, providing a reference for the formulation of the vacuum carburizing process for practical production and application. Li, X. et al. [18] conducted the numerical simulation and experimental study of the actual carburizing process of helical gears and verified the phase transformation properties and the influence on the distortion and residual stresses after carburizing and quenching. Su, J. et al. [19] proposed the heat distribution ratio and convective cooling coefficient of the grinding process and conducted the finite element simulation of grinding, showing that the change of tooth surface temperature was generated through the grinding process. Tang, H. et al. [20] conducted finite element simulation of the forging processes, analyzed the effects of initial forging temperature and passes on the temperature and strain rate of forgings, and obtained the suitable forging temperature of 1000–1100 °C in the forging passes.

In summation, scholars have studied the processing technology of 20CrMnTi or conducted research on the metamorphic layer and work hardening behavior formed in the process of metal cutting. However, there are currently no papers in the literature dedicated to the study of the metamorphic layer and work hardening aspects of contour bevel gears. Therefore, in order to control the surface’s metamorphic layer and improve the performance of the workpiece, the finite element simulation analysis of the single tooth of the contour bevel gears is carried out in this paper. The temperature field, strain field, and strain rate at different depths from the machined surface under different cutting parameters are extracted. The machined hardening degree of the surface’s metamorphic layer under different cutting parameters is studied by performing a cutting test on contour bevel gears, and the diffraction analysis of the machined metamorphic layer is carried out using XRD to explore the variation of grain refinement and material phase transformation. The effect of cutting parameters on the thickness of the machined surface’s metamorphic layer is studied using a scanning electron microscope. The research results provide an experimental basis for gear surface quality optimization, improving the smoothness of gear meshing transmission and improving the fatigue strength of the workpiece.

## 2. Materials and Methods

### 2.1. Finite Element Simulation Model of Milling Contour Bevel Gears

#### 2.1.1. Material Constitutive Model

As a commonly used dynamic flow stress model, the Johnson-Cook model takes into account the post-yielding strengthening and temperature rise softening effects of the material, which is consistent with the large plastic deformation as well as the thermal-mechanical coupling of high strain rate and the rapid accumulation of cutting heat, which cause the material to soften. In this paper, the Johnson-Cook model will be used, and its expression is:(1)$$\sigma_{p}=[A+B{\left(\varepsilon_{p}\right)}^{n}]\left[1+C\ln(\frac{\dot{\varepsilon }}{{\dot{\varepsilon }}_{0}}\right)][1-\left(\frac{T-T_{r}}{T_{m}-T_{r}}{)}^{m}\right]$$

In the formula, $\sigma_{p}$ is the flow stress, $\varepsilon_{p}$ is the plastic strain, $\dot{\varepsilon }$ is the strain rate, ${\dot{\varepsilon }}_{0}$ is the reference strain rate, T is the experimental temperature, $T_{m}$ is the material melting point, $T_{r}$ is the reference temperature, A is the yield criterion under quasi-static conditions, B is the strain hardening parameter, C is the strain rate strengthening parameter, n is the processing Hardening index, and m is the softening parameter.

Johnson-Cook model of 20CrMnTi and basic physical properties of 20CrMnTi are shown in Table 1 and Table 2.

#### 2.1.2. Tool-Chip Contact Conditions

The milling process of the contour bevel gears is multi-edge intermittent cutting. The chips do not have the lubricating and cooling effect of the cutting fluid in the process of contacting the cutter teeth, and there is a large friction force. The thermal-mechanical coupling in the cutting process will directly affect the plastic deformation process of the material. Therefore, the modified Coulomb friction model as the tool-chip contact model is selected in this paper, represented in Formula (2):(2)$$f=\{\begin{array}{l} \mu \cdot \sigma_{n}\;\left(\mu \cdot \sigma_{n}\leq \tau_{s}\right)\\ \tau_{s}\;\;\;\;\;\;\;\left(\mu \cdot \sigma_{n}\geq \tau_{s}\right) \end{array}$$

In the formula, f is the friction force, μ is the friction coefficient between the chip and the rake face, $\sigma_{n}$ is the normal stress of the cutter-chip interface, $\tau_{s}$ is the shear yield stress of the material, and m is the coefficient.

#### 2.1.3. Cutting Heat Conduction Equation

The cutting heat during the milling of the contour bevel gears mainly comes from the plastic deformation of the workpiece in the first deformation zone and the friction between the rake face and the chip in the second deformation zone. The cutting deformation area is shown in Figure 1.

The conduction equation of the cutting heat in the first deformation zone:(3)$$K\frac{\partial^{2}T}{\partial x^{2}}+\dot{Q}=\rho C_{p}\left(u\frac{\partial T}{\partial x}+v\frac{\partial T}{\partial y}\right)$$
(4)$$\dot{Q}=k\sigma_{ij}\varepsilon_{i}{}_{j}$$

In the formula, K is the heat transfer coefficient, k is the heat conversion coefficient, ρ is the material density, $C_{p}$ is the material specific heat, $\sigma_{ij}$ is the material stress, $\varepsilon_{ij}$ is the material strain rate, and $\dot{Q}$ is the heat generation rate of the conversion of mechanical energy and thermal energy per unit volume.

The equation of heat generated by friction in the second deformation zone is:(5)$$Q=\tau_{f}v_{f}$$

In the formula, $\tau_{f}$ is the shear stress between the rake face and the chip and $v_{f}$ is the relative velocity between the rake face and the chip.

#### 2.1.4. Failure Separation Criteria

This paper adopts the Cockcroft and Latham fracture criterion built in the cutting simulation software, DEFORM software.
(6)$$\sigma^{*}=\{\begin{array}{l} \sigma_{1}\\ 0 \end{array}\begin{array}{l} \sigma_{1}\geq 0\\ \sigma_{1}\leq 0 \end{array}\;,\;\int_{0}^{{\bar{\varepsilon }}_{f}}\sigma^{*}d\bar{\varepsilon }=C$$

In the formula, $\sigma^{*}$ is the equivalent stress, $\bar{\varepsilon }$ is the equivalent strain, ${\bar{\varepsilon }}_{f}$ is the equivalent plastic strain at fracture, $\sigma_{1}$ is the maximum value among the three principal stresses, and C is the fracture threshold of each criterion.

#### 2.1.5. Simulation Model

The milling process of the contour bevel gears is multi-blade intermittent cutting. The gear blank and the cutterhead rotate according to a certain transmission ratio. The gear forming process is related to the geometric structure of the blade and the relationship of the generating motion. The mutual movement relationship between the tool blade and the gear blank is shown in Figure 2, which is the coordinate system of the gear and the cutterhead.

The model is parametrically modeled using the data of the tool blade of the contour bevel gears, and only the part of the cutter teeth with the cutting edge that participates in cutting is retained for simplification, as shown in Figure 3. 

The blank of the contour bevel gear is converted into the blank of the imaginary generating gear to improve the calculation accuracy, and then the simplified blank is used as the simulation workpiece model, as shown in Figure 4.

The tool grid is set to 32,000, the grid number of the workpiece is 100,000, the minimum grid size of the tool is 0.1 mm, and the minimum grid size of the workpiece is 0.08 mm. For local division, the division ratios of local meshes are 0.1 and 0.5, respectively, as shown in Figure 5.

#### 2.1.6. Selection of Simulation Parameters

The cutting parameters are mainly selected according to the actual processing conditions which is shown in Table 3.

### 2.2. Dry Milling Test Preparation of Contour Bevel Gears

#### 2.2.1. Test Conditions and Materials

The cutting test of the contour bevel gear is carried out on a Phoenix 175 HC CNC machine, made by Gleason Corporation in Rochester, New York, NY, USA, as shown in Figure 6C. Figure 6B shows the blades for machining contour bevel gears. It is divided into internal and external blades. The rake angle of the blades is 12°, the tool clearance is 19°34′, the conner radius of the main cutting edge is 1.52 mm, the tool point width is 3.45 mm. and the main pressure angle is 21°19′. It is assembled with the cutterhead to form a disc cutter, and the maximum cylindrical diameter of the cutter is 320 mm. A TRI-AC- cutterhead and DT-270D3-39/8-PN blade are used in the test, as shown in Figure 6A. Figure 6D is the gear blank of the contour bevel gear. The gear blank is normalised at 860°, cooled in air to 600°, held for 8 h and then air cooled to 25°. By clamping the disc cutter and the blank on the machine tool and adjusting the installation angle, the milling of the contour bevel gear can be processed. Figure 6E is the machined contour bevel gear. The chemical composition of the workpiece was examined by energy spectrum analysis and the results are shown in Table 4.

#### 2.2.2. Experiment Programme Design

This paper used the generating method to machine the contour bevel gears. Therefore, the workpiece and the tool undergo a generating movement. The milling process usually adopts a one-time feed mode, and the main cutting parameters are cutting speed and feed rate. Therefore, the internal relationship between the cutting speed and feed rate of the tool and the characteristics of the machined metamorphic layer is studied respectively, and the appropriate cutting parameters are determined based on the above simulation results. The specific experimental design is shown in Table 5.

#### 2.2.3. Preparation of Metallographic Specimens

The tooth blank is processed into the contour bevel gear through the milling test, and the metallographic sample with a length of 10 mm is extracted by slow wire cutting, then the oil on the machined surface is cleaned ultrasonically, as shown in Figure 7. The samples are fixed by hot inlay method, and the inlay material is metallographic inlay powder. Water sandpaper is used to smooth the section of the sample to be tested, and the mesh numbers of the water sandpaper are 200, 360, 500, 800, 1000, 1500, and 2000 in sequence. Then, the ground sample is mechanically polished. Finally, chemical etching is carried out on the metallographic sample, and the etching agent is 4% nitric acid alcohol solution.

## 3. Results and Discussion

### 3.1. Simulation Results and Analysis

The whole process of single-tooth milling simulation is performed by DEFORM, as shown in Figure 8.

Under the condition of high temperature and high strain, the grain structure of 20CrMnTi steel will undergo different degrees of phase transformation and crystallization behavior. In the process of milling the contour bevel gears, the temperature field and deformation field generated during the cutting process are directly related. Microscopic evolution to machined surfaces causes formation of metamorphic layers. Therefore, the influence of the cutting parameters on the temperature field and deformation field is simulated and analyzed, and the temperature, stress and strain energy of the machined section of the workpiece are extracted, which will pave the way for the subsequent experimental research of the metamorphic layer and the reasonable selection of cutting parameters.

The milling cutterhead is assembled by internal and external blades. Since the generating motion of the internal and external blades is the same, this model mainly studies the condition of the outer cutter while machining the concave surface of the gear. Since the cutting part is mainly the side of the blade, the maximum temperature and stress of the machined section of the workpiece are located at the contact position between the side of the tool and the workpiece. Data such as temperature and deformation of the microstructure of the machined section are extracted along the direction perpendicular to the cutting section [5], as shown in Figure 9.

Figure 10 shows the variation of cutting temperature and strain with the depth of the machined surface under the cutting conditions of v = 170 m/min and f = 0.1 mm/r. Figure 10A,B are the cutting temperature and strain under the cutting profile, respectively. Figure 10C shows the variation of cutting temperature and strain with the depth of the machined surface. Figure 10 shows that as the depth from the machined surface increases, both the cutting temperature and the strain decrease, but the slope of the strain curve is greater than that of the cutting temperature, indicating that the strain changes more significantly than the cutting temperature.

Figure 11A–C respectively show the diagrams and partial diagrams of the influence of the feed rate on the strain field, strain rate, and temperature field of the machined surface. The temperature field of the machined surface is increased, and strain and strain rate fields both tend to increase.

Figure 12A–C respectively show the diagrams and partial diagrams of the effect of cutting speed on the strain field, strain rate, and temperature field of the machined surface. With the increase in cutting speed, the work per unit time increases, and the temperature field of the machined surface increased. When *v* = 200 m/min, the temperature is the highest, then gradually decreases. The strain and strain rate gradually increased, but not as much as the feed rate.

### 3.2. Analysis of Test Results

#### 3.2.1. XRD Diffraction Analysis

In order to explore the characteristics of the machined surface’s metamorphic layer, the machined metamorphic layer is analyzed by XRD diffraction, and the internal relationship between cutting parameters and metamorphic layer characteristics is explored. The basic parameters of X-ray diffraction test are shown in Table 6.

Since the diffraction peak of the processed metamorphic layer of 20CrMnTi is mainly concentrated in 43°~46°, the characteristics of the machined metamorphic layer are explored through the diffraction peak in this range.

Figure 13 shows that due to the increase in cutting speed, the diffraction intensity and width are increasing, resulting in grain refinement. When the cutting speed is increased to 230 m/min, the diffraction peak is close to the speed of 200 m/min. Combined with the simulation analysis of cutting temperature, it can be speculated that the machined surface temperature of the workpiece is increased because of the increase in cutting speed, causing it to reach the phase transition temperature, and the substrate material generates the phase transition. When the cutting speed continues to increase, the cutting temperature and the increase rate of diffraction peak are reduced. It can be found that ferrite mainly exists in the form of α-Fe and γ-Fe by analyzing the diffraction peaks.

Figure 14 shows that the diffraction peak of the machined metamorphic layer gradually increases with the increase in the feed rate, resulting in grain refinement and increasing the hardness of the machined metamorphic layer.

#### 3.2.2. Observation of Metallographic Morphology

Figure 15A,B are respectively the metallographic morphology of the machined metamorphic layer with the feed rate of 0.1 mm/r and 0.2 mm/r at the cutting speed of 260 m/min. It shows that the substrate material of the machined section comprises ferrite and pearlite. During the machining process, high temperature and strain are produced in the cutting area, which causes some ferrite and pearlite to transform into austenite. Under XRD diffraction, grain refinement can be observed. It shows that there is a metamorphic layer in the machined section, which leads to work hardening of the machined section and affects the machined surface quality.

#### 3.2.3. Analysis of Work Hardening

The machined contour bevel gears are cut along the direction perpendicular to the machined surface by wire cutting equipment, and the distribution law of hardness in the depth direction from the machined surface is explored by means of cross-section marking. The specific parameter settings for the hardness tester are shown in Table 7. 

In order to prevent the influence between adjacent indentations, 10 test points are taken along the distance direction of the machined surface, and the indentation spacing is three times the length of the indentation diagonal. Three hardness tests are carried out at the same depth, and the average value of the three tests is taken as the hardness value of the depth position to reduce the test error. The depth spacing is set as shown in Figure 16 [21].

Figure 17 shows that when the cutting speed increases from 170 m/min to 200 m/min, the plastic deformation speed of the workpiece increases, the first deformation zone becomes narrower, and the degree of work hardening becomes larger. When the cutting speed rises to 230 m/min, because the cutting time is greatly shortened, the cutting heat has no time to dissipate, resulting in an enhanced thermal softening effect and a decrease in work hardening. However, it can be found that with the increase in cutting speed, the change trend of work hardening is not particularly obvious, which may be caused by the double effect of the increase in cutting speed on work hardening and the uneven hardness distribution of the workpiece material.

In combination with Figure 18, it shows that the increase in the feed rate causes the increase in the hardness of the metamorphic layer on the machined surface. This is because distortion of the lattice and deformation resistance increase with the increase in the feed rate, and the degree of work hardening of the metamorphic layer increases.

Comparing the simulation data with the test data at *f* = 0.1 mm/r, *v* = 260 m/min, it is found that when the depth from the machined surface is between 0 and 40 μm, the hardness of the metamorphic layer changes greatly, as shown in the area on the left side of the dashed line in Figure 19. The transformation amplitude tends to be gentle at 40~90 μm, as shown in the area on the right side of the dashed line of Figure 19, which is the same as the change trend of the above simulation strain and temperature, thus verifying the accuracy of the finite element simulation.

#### 3.2.4. EDS Spectrum Analysis of Metamorphic Layer

It can be seen from the above analysis that, during the cutting process performed on the contour bevel gears, there is a metamorphic layer in the section of the machined workpiece, which affects the machined surface quality. The machined sections are mainly divided into three regions: metamorphic zone (A), transition zone (B), and substrate zone (C), as shown in Figure 20. Therefore, EDS is used to analyze the energy spectrum of the three regions to detect the change of element content during the formation of the metamorphic layer.

The energy spectrum analysis of different regions of metamorphic layer is shown in Figure 21. It can be found that there are Fe, Cr, and Mn elements in the metamorphic layer region, transition region, and substrate region, and the element contents are different. The closer to the metamorphic layer region, the lower the Fe element content is. However, there are few kinds of elements measured by energy spectrum analysis, which may be due to the loss of some elements during the process of metallographic corrosion. 

Combined with XRD phase analysis, the energy spectrum analysis also shows that in the cutting process, the temperature of the cutting area gradually increases, and the local microstructure of the substrate material generates phase transformation, resulting in the metamorphic layer area; while the transition area is the area where the metamorphic layer comes into contact with the substrate material, which causes some ferrite and pearlite to transform into austenite, consistent with the above XRD diffraction analysis results.

#### 3.2.5. Scanning Electron Microscopy Analysis of Metamorphic Layer

The above analysis shows that the microstructure and element content of the metamorphic layer region and the substrate region of the contour bevel gears are different. Therefore, the thickness of the metamorphic layer at three different positions of the metallographic specimen of each contour bevel gears is measured by scanning electron microscopy. After taking the average value, the influence of the cutting parameters on the thickness of the metamorphic layer of the contour bevel gears is explored.

Figure 22 is a line graph of the variation of the thickness of the metamorphic layer in regards to the feed rate when the cutting speed is 260 m/min. With the increase in the feed rate, the thickness of the machined metamorphic layer of the contour bevel gears increases. Combined with the simulation analysis of Figure 9A, it can be found that the temperature of the cutting area increases with the increase in the feed rate, which leads to the phase change of the workpiece material. In addition, the increase in the feed rate will lead to an increase in the thickness of the chip, which will squeeze the workpiece. Therefore, the thickness of the metamorphic layer increases.

Figure 23 is a line graph of the variation of the thickness of the metamorphic layer in regards to the cutting speed when the feed rate is 0.1 mm/r. It can be found that with the increase in cutting speed, the thickness of the machined metamorphic layer of the contour bevel gears increases gradually. From the simulation analysis in Figure 10A,B, it can be seen that the increase in cutting speed can cause an increase in surface strain and strain rate, resulting in the grain refinement of the metamorphic layer and increasing the thickness of the metamorphic layer.

## 4. Conclusions

In this paper, the characteristics of the machined metamorphic layer and the work hardening behavior of milling contour bevel gears are studied. The influence of the cutting parameters on the work hardening and the machined metamorphic layer of milling contour bevel gears is explored via single-tooth cutting simulation and contour bevel gears cutting tests. Main results are summarized in the following.

(1)Based on finite element simulation and heat transfer theory, the milling process of contour bevel gears is simulated. The temperature field, strain field, and strain rate of metamorphic layer at different depths from the machined surface are extracted and analyzed. It is found that with an increase in feed rate, the temperature field of the machined surface of contour bevel gears increases, and the strain and strain rate fields tend to increase. With an increase in cutting speed, the work per unit time increases, and the temperature field of the machined surface increases gradually. As the depth of the machined surface increases, the rate of change of the temperature, strain, and strain rate decreases and eventually stabilizes.(2)Diffraction analysis of the metamorphic layer of contour bevel gears by XRD revealed that α-Fe and γ-Fe mainly exist in the metamorphic layer. An increase in cutting speed causes an increase in diffraction intensity and width, resulting in grain refinement. Super-depth observation of metallographic specimen is carried out to verify the existence of the metamorphic layer on the cutting surface. It is found that the substrate material on the machined surface comprises mainly ferrite and lamellar pearlite, and a small amount of austenite exists in the metamorphic layer area. The work hardening test of the machined metamorphic layer shows that the hardness of the metamorphic layer increases first and then decreases with an increase in cutting speed, and also increases with an increase in the feed rate. When the depth from the machined surface is 0~40 μm, the machined section is located in the metamorphic zone and the transition zone, and the hardness of the metamorphic layer changes greatly; whereas when the depth is 40~90 μm, the machined section is located in the transition zone and the substrate material zone, and the change tends to be gentle and gradually close to the substrate hardness.(3)It is found that a metamorphic zone, transition zone and substrate zone exist in the machined section of gear. EDS is used to analyze the energy spectrum of different regions of the metamorphic layer, and it is found that the content of Fe element in the region near the metamorphic layer is low, indicating that there is a phase transition in the metamorphic layer region. Combined with finite element simulation and XRD diffraction analysis, it is found that the high temperature and strain generated in the cutting area transforms part of the ferrite and pearlite of the matrix into austenite, resulting in an increase in hardness of the metamorphic layer area, which verifies the accuracy of the work hardening test of the metamorphic layer.(4)The thickness of the metamorphic layer of the contour bevel gears with different cutting parameters is observed, and it is found that the thickness of the machined metamorphic layer increases with an increase in the cutting speed and the feed rate. When the cutting speed is in the range of 200~230 m/min and the feed rate is in the range of 0.1~0.2 mm/r, the change in the thickness and the error bar of the metamorphic layer is smaller than other layers. Therefore, milling contour bevel gears in this parameter range can effectively control the thickness of the metamorphic layer, providing a basis for improving the smoothness of gear meshing transmission and improving the fatigue strength of the workpiece.

## Figures and Tables

**Figure 1 materials-15-07975-f001:**
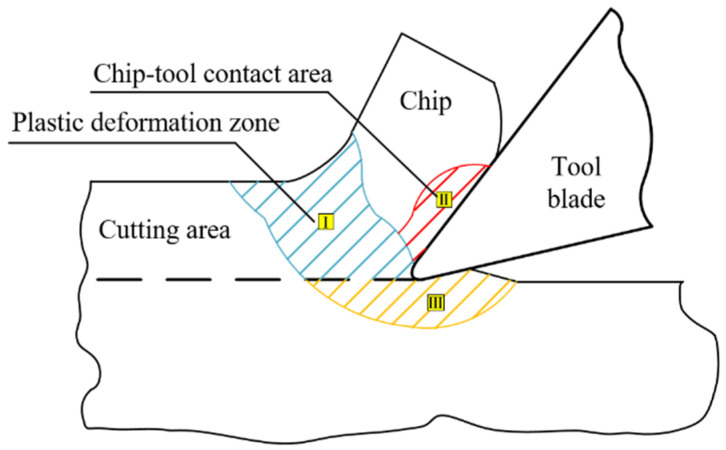
Cutting deformation area of contour bevel gears.

**Figure 2 materials-15-07975-f002:**
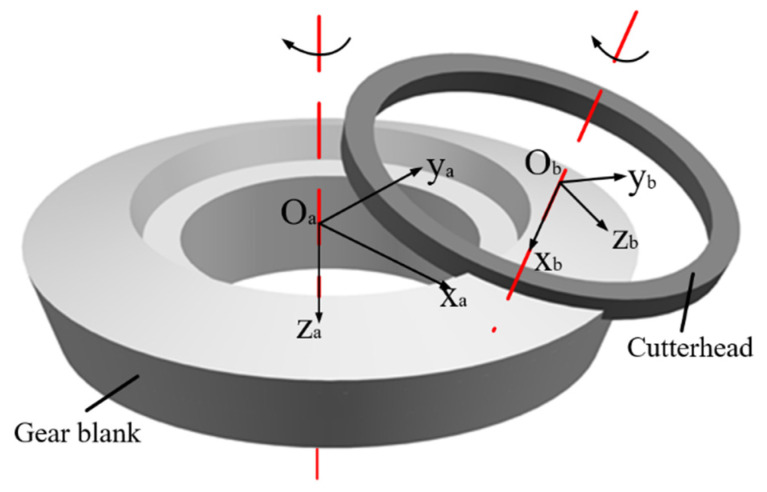
Establishment of the motion relationship model.

**Figure 3 materials-15-07975-f003:**
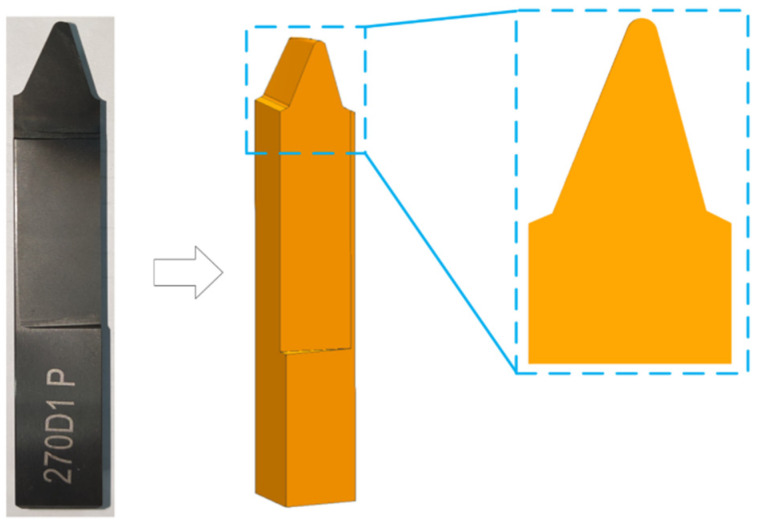
Simplified blade model.

**Figure 4 materials-15-07975-f004:**
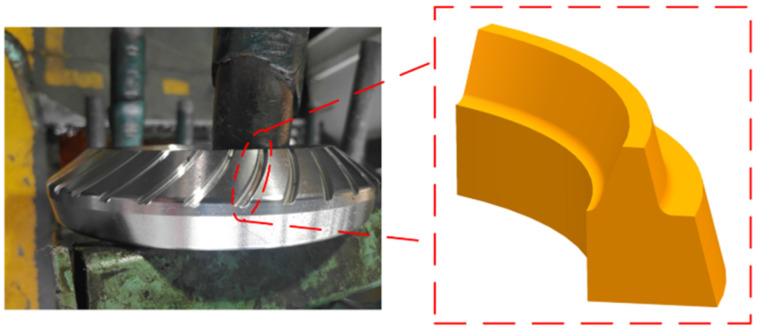
Simplified workpiece model.

**Figure 5 materials-15-07975-f005:**
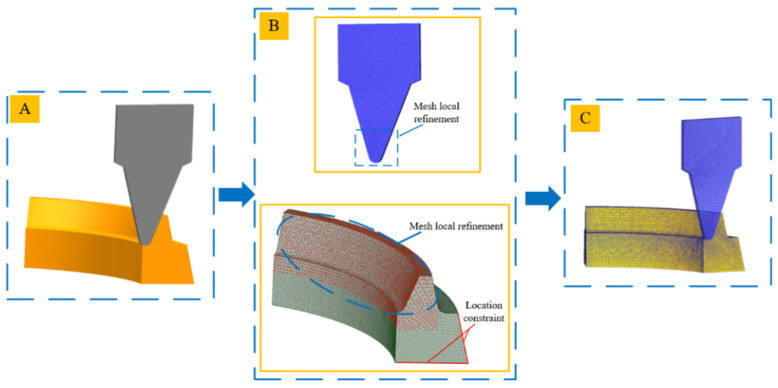
Assembly model and meshing ((**A**) shows the assembly model, (**B**) shows the process of mesh refinement, and (**C**) shows the result of mesh refinement).

**Figure 6 materials-15-07975-f006:**
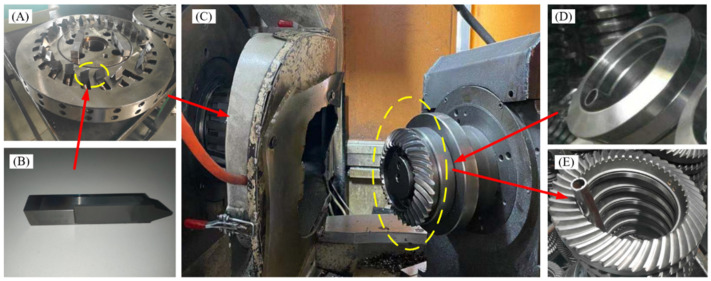
Dry milling machine device for contour bevel gears. ((**A**) shows the TRI-AC cutterhead, (**B**) shows the blades, (**C**) shows the Phoenix 175 HC CNC machine, (**D**) shows the gear blank, and (**E**) shows the machined contour bevel gear).

**Figure 7 materials-15-07975-f007:**
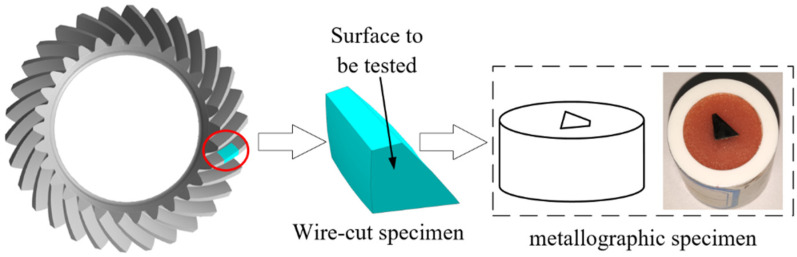
Schematic diagram of metallographic specimen production.

**Figure 8 materials-15-07975-f008:**
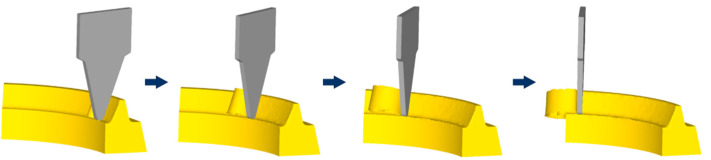
The whole process of single tooth cutting.

**Figure 9 materials-15-07975-f009:**
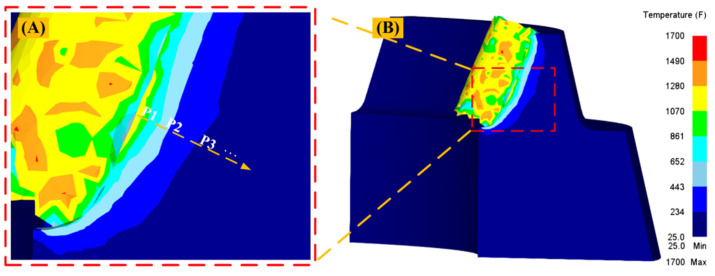
Schematic diagram of extraction of cutting temperature field, strain rate, and strain. (**A**) The temperature field of machined section. (**B**) The direction of data extraction of temperature and deformation of machined sections.

**Figure 10 materials-15-07975-f010:**
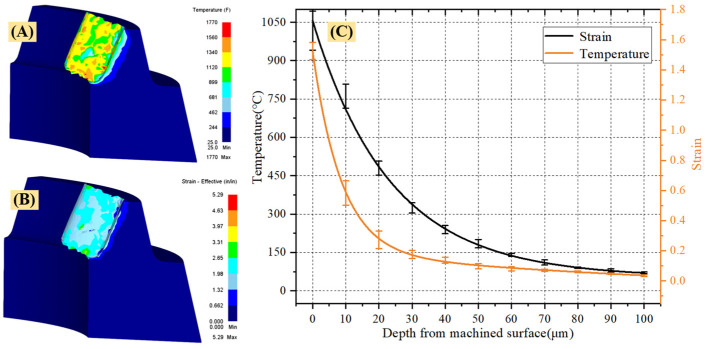
(*v* = 170 m/min, *f* = 0.1 mm/r) variation of cutting temperature and strain with depth of machined surface ((**A**) shows the temperature field of the cutting section, (**B**) shows the strain field of the cutting section, and (**C**) shows the variation of cutting temperature and strain with the depth of the machined surface).

**Figure 11 materials-15-07975-f011:**
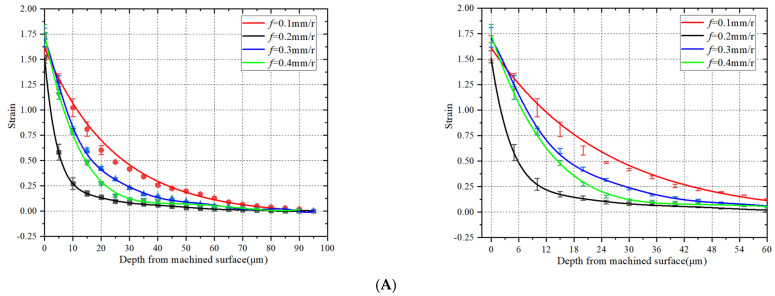
Effect of feed rate on strain, strain rate and temperature of machined surface. (**A**) Effect of feed rate on strain of machined surface. (**B**) Effect of feed rate on strain rate of machined surface. (**C**) Effect of feed rate on temperature of machined surface.

**Figure 12 materials-15-07975-f012:**
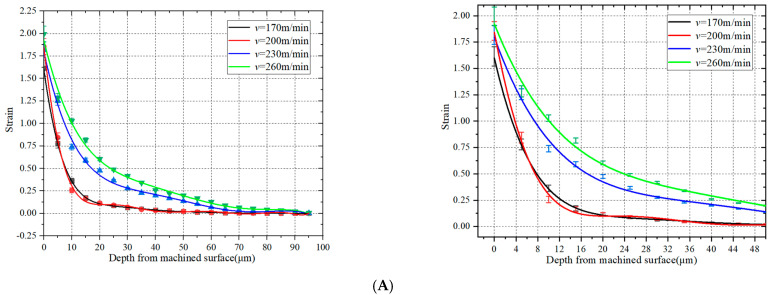
Effect of cutting speed on strain, strain rate, and temperature of machined surface. (**A**) Effect of cutting speed on strain of machined surface. (**B**) Effect of cutting speed on strain rate of machined surface. (**C**) Effect of cutting speed on temperature of machined surface.

**Figure 13 materials-15-07975-f013:**
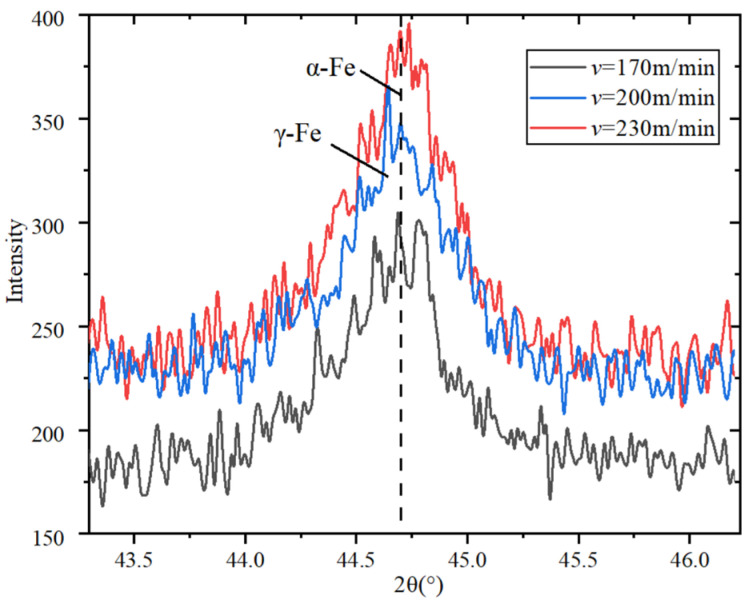
Effect of cutting speed on diffraction peak of metamorphic layer.

**Figure 14 materials-15-07975-f014:**
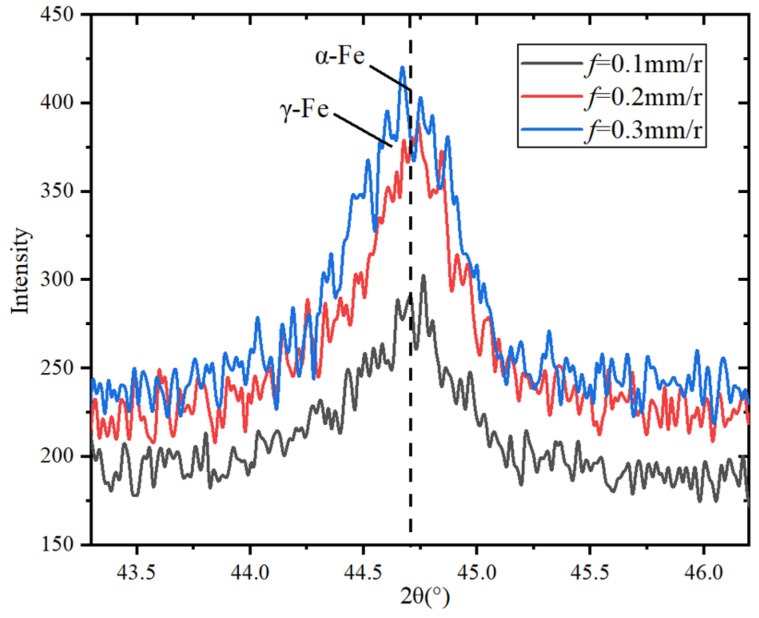
Effect of feed rate on diffraction peak of metamorphic layer.

**Figure 15 materials-15-07975-f015:**
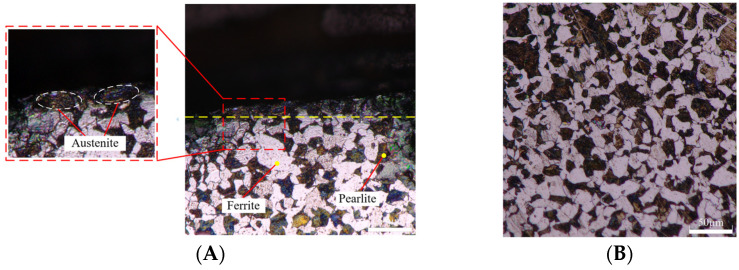
Metallographic morphology of metamorphic layer and surface. (**A**) Metallographic morphology of metamorphic layer. (**B**) Metallographic morphology of surface.

**Figure 16 materials-15-07975-f016:**
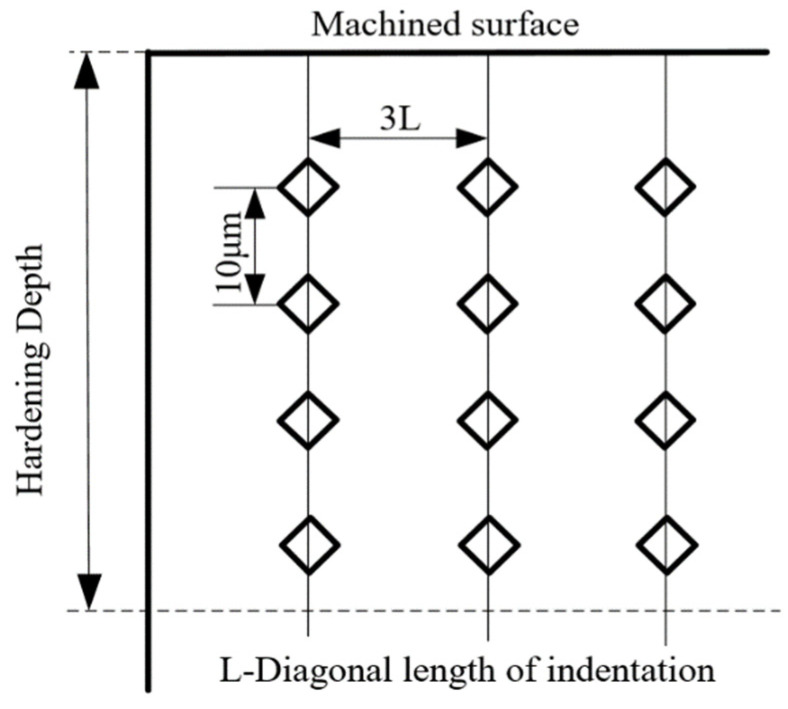
Hardness measurement schematic diagram.

**Figure 17 materials-15-07975-f017:**
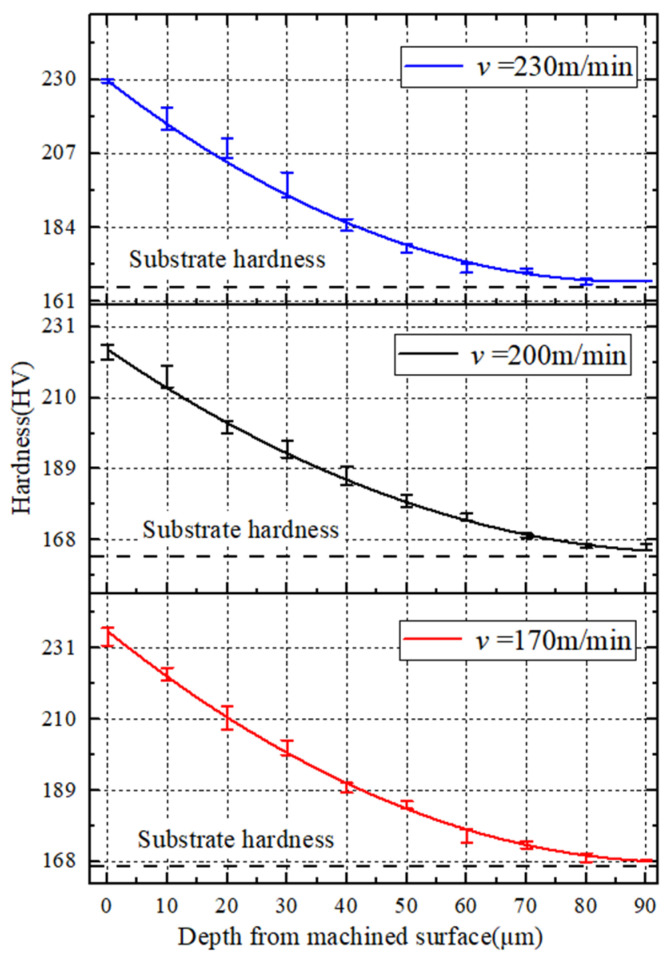
(*f* = 0.1 mm/r) Influence of cutting speed on work hardening.

**Figure 18 materials-15-07975-f018:**
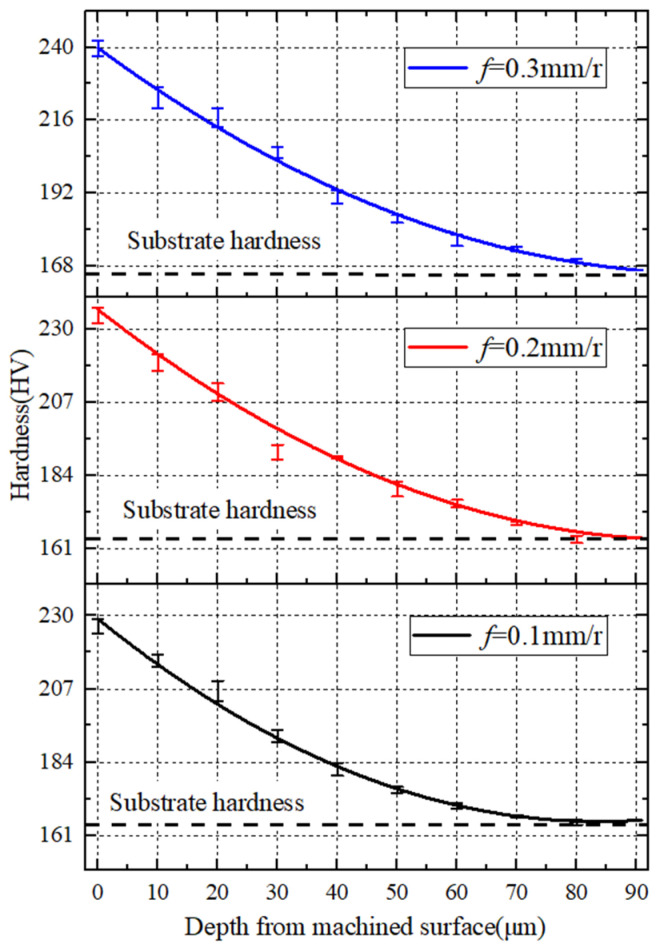
(*v* = 260 m/min) Influence of feed rate on work hardening.

**Figure 19 materials-15-07975-f019:**
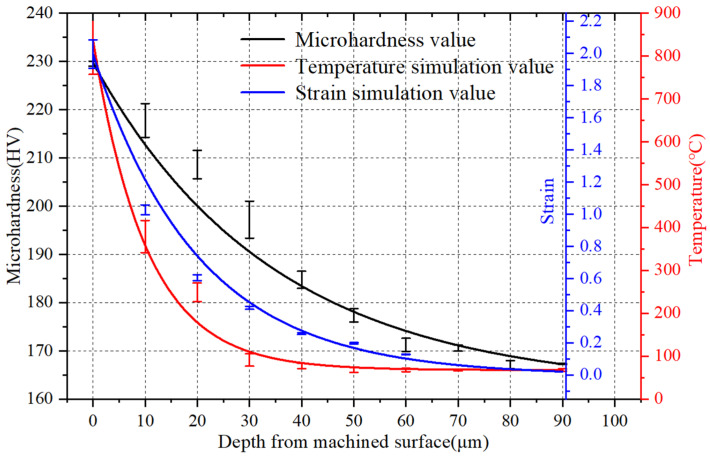
Comparative analysis of simulation and test.

**Figure 20 materials-15-07975-f020:**
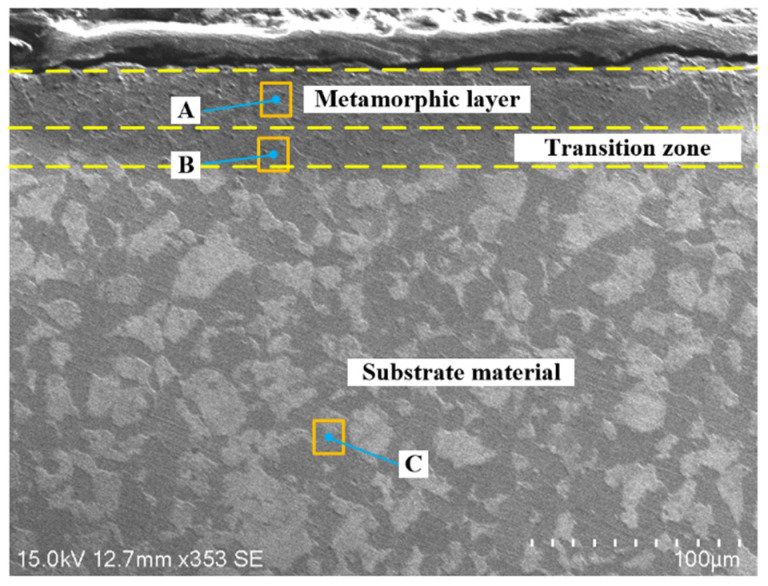
EDS detection area of metamorphic layer.

**Figure 21 materials-15-07975-f021:**
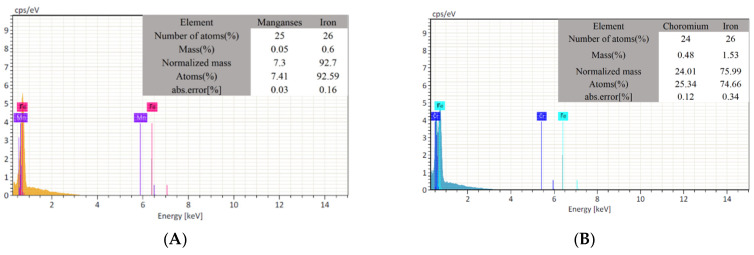
EDS spectrum analysis results of metamorphic layer. (**A**) EDS spectrum analysis of metamorphic layer; (**B**) EDS spectrum analysis of transition zone; (**C**) EDS spectrum analysis of substrate material.

**Figure 22 materials-15-07975-f022:**
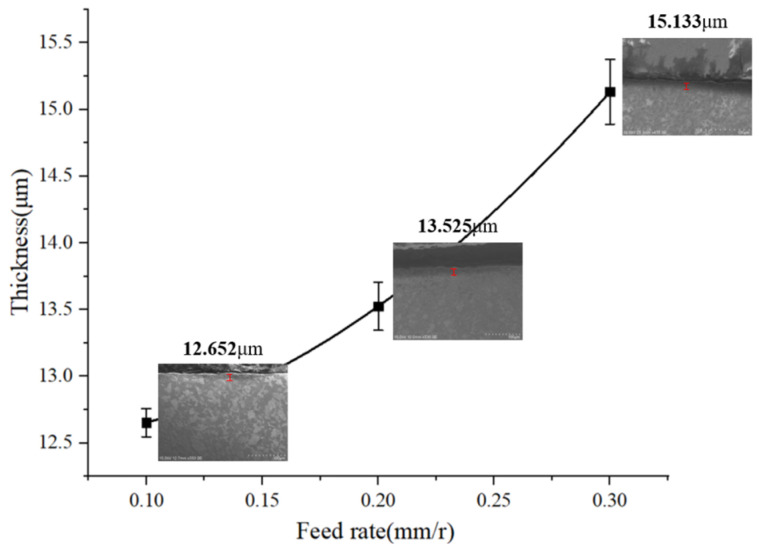
Variation of metamorphic layer thickness with feed rate.

**Figure 23 materials-15-07975-f023:**
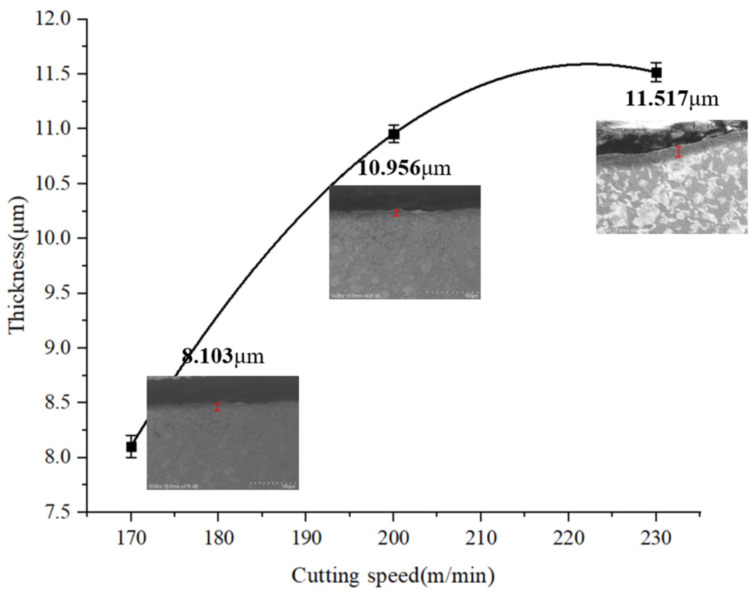
Variation of metamorphic layer thickness with cutting speed.

**Table 1 materials-15-07975-t001:** Johnson-Cook model of 20CrMnTi.

Yield Strength *A* (Mpa)	Hardening Modulus B (Mpa)	Hardening Coefficient n	Strain Rate Coefficient C	Thermal Softening Coefficient m
1241	622	0.6522	0.0134	1.3

**Table 2 materials-15-07975-t002:** Basic physical properties of 20CrMnTi.

Density *ρ* (kg·m^3^)	Elastic Modulus *E* (Gpa)	Poisson Ratio *μ *	Hardness (HV)
8270	205	0.3	≤217

**Table 3 materials-15-07975-t003:** Selection of simulation cutting parameters.

Cutting Parameters	Cutting Speed *ν* (m/min)	Feed Rate *f* (mm/r)
1	200	0.1, 0.2, 0.3, 0.4
2	170, 200, 230, 260	0.1

**Table 4 materials-15-07975-t004:** Chemical constituents of 20CrMnTi.

Element	C	Si	Mn	Cr	S	P	Ni	Ti
Content (%)	0.17–0.23	0.17–0.37	0.80–1.10	1.00–1.30	0.007–0.015	0.01–0.02	0.003–0.004	0.04–0.10

**Table 5 materials-15-07975-t005:** Selection of Test Cutting Parameters.

Cutting Parameters	Cutting Speed *ν* (m/min)	Feed Rate *f* (mm/r)
1	200	0.1, 0.2, 0.3
2	170, 200, 230	0.1

**Table 6 materials-15-07975-t006:** Basic parameters of X-ray diffraction test.

Divergence Slit	Diffuse Slit	Collector Slit	Scan Speed	Sampling Step Width
1°	1°	0.3 mm	4°/min	0.02 (2θ)

**Table 7 materials-15-07975-t007:** Specific parameters of Vickers hardness tester.

Model Type	Load F (N)	Holding Time T (s)
DHV-1000	0.98	15

## Data Availability

The data presented in this study are available from the corresponding author upon reasonable request.

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
