# Peer review of "Study on the Work Hardening and Metamorphic Layer Characteristics of Milling Contour Bevel Gears"

_materials, 2022, doi:10.3390/ma15227975_

Round 1

Reviewer 1 Report

The article has a scientific character. The article deals with the formation of metamorphic layers when dry milling taper wheels. The Authors applied correct research methods and used the appropriate measuring equipment. The content of the work is logically written. The manuscript contains 24 figures and 5 tables. Figures and tables aren’t properly prepared. Authors cited 16 literature sources. The authors presented an interesting work, but it requires numerous improvements to be of satisfactory quality.

General remarks

1. Specify the data of the apparatus in the article in the following order: device designation, manufacturer's name, city, country.

2. In general, improve the discussion of the results obtained

3. The work also requires careful text editing

Detailed comments

1. Line 183 Give the method of determining the chemical composition; provide also the type and conditions of heat treatment of the wheel prefabricated element.

2. In Figure 5, 6 in the description below the figure, give what is shown in figures A ÷ E

3. Table 4 Give the measurement uncertainty in the table

4. Verse 188 Give the method of cutting the teeth (Gleason introduced many methods),

5. Figure 10 ÷ 12, 18 ÷ 20, 23, 24 in the figures, do not connect the results with segments, but enter the polynomial regression lines; provide error bars in the figure

6. Line 258 Describe in detail the XRD test conditions in Chapter 2

7. Figure 13 The authors must also show the changes of the peaks after milling in the area of ​​the austenite phase (γ - Fe)

8. Figure 15 in the picture does not show austenite, give a better picture; there should also be a photo with the output structure without a metamorphic layer

9. Figure 16 figure not needed, delete it

10. Figure 22 not legible, Authors must correct it

11. Line 287 "ferrite and pearlite transform into austenite" - explain the process precisely, it is important because of the topic of work

Additional questions:

- as a result of high temperature, cracks in the top layer may form, have you observed such damage?

- what was the surface roughness after milling under different conditions?

The answers to these questions should also be included in the text

Author Response

Response to Reviewer 1 Comments

Point 1: Specify the data of the apparatus in the article in the following order: device designation, manufacturer's name, city, country.

Response 1: Thanks to the reviewers for your comments, which have been revised as requested and are highlighted in red on line 191 of the manuscript. The modified sentence is expressed as follows:

The cutting test of the contour bevel gear is carried out on a Phoenix 175 HC CNC machine, made by Gleason Corporation in the USA.

Point 2: In general, improve the discussion of the results obtained.

Response 2: Thanks to the reviewers for your comments, the discussion of the results of this paper has been carefully sorted and revised, and are marked with red fonts on line 407 of the revised manuscript.

Point 3.1: Line 183 Give the method of determining the chemical composition; provide also the type and conditions of heat treatment of the wheel prefabricated element.

Response 3.1: Thanks to the reviewers for your comments, which have been revised as requested and are highlighted in red on line 199 and 202 of the manuscript. The modified sentence is expressed as follows:

The chemical composition of the workpiece was examined by energy spectrum analysis and the results are shown in Table 4.

The gear blank is normalised at 860°, cooled in air to 600°, held for 8h and then air cooled to 25°.

Point 3.2: In Figure 5, 6 in the description below the figure, give what is shown in figures A ÷ E.

Response 3.2: Thanks to the reviewers for your comments, which have been revised as requested and are highlighted in red in Figure5 and Figure 6 of the manuscript. The modified sentence is expressed as follows:

Figure 5. Assembly model and meshing (Figure A shows the assembly model, Figure B shows the process of mesh refinement and Figure C shows the result of mesh refinement)

Figure 6. Dry milling machine device for contour bevel gears (Figure A shows the Tri-ac cutter head, Figure B shows the blades, Figure C shows the Phoenix 175 HC CNC machine, Figure D shows the gear blank and Figure E shows the machined contour bevel gear)

Point 3.3: Table 4 Give the measurement uncertainty in the table.

Response 3.3: Thanks to the reviewers for your comments, which have been revised as requested and are highlighted in red in Table 4 of the manuscript.

Point 3.4: Verse 188 Give the method of cutting the teeth (Gleason introduced many methods)

Response 3.4: Thanks to the reviewers for your comments, which have been revised as requested and are highlighted in red on line 210 of the manuscript. The modified sentence is expressed as follows:

This paper used the generating method to machine the contour bevel gears. Therefore, the workpiece and the tool undergo a generating movement.

Point 3.5: Figure 10 ÷ 12, 18 ÷ 20, 23, 24 in the figures, do not connect the results with segments, but enter the polynomial regression lines; provide error bars in the figure.

Response 3.5: Thanks to the reviewers for your comments, which have been revised as requested and are highlighted in red at the relevant places in the manuscript.

Point 3.6: Line 258 Describe in detail the XRD test conditions in Chapter 2

Response 3.6: Thanks to the reviewers for your comments, which have been revised as requested and are highlighted in red in Table 7 of the manuscript.

Point 3.7: Figure 13 The authors must also show the changes of the peaks after milling in the area of the austenite phase (γ - Fe).

Response 3.7: Thanks to the reviewers for your comments, which have been revised as requested and are highlighted in red in Figure 13,Figure 14 and the conclusion of the manuscript.

Point 3.8: Figure 15 in the picture does not show austenite, give a better picture; there should also be a photo with the output structure without a metamorphic layer.

Response 3.8: Thanks to the reviewers for your comments, which have been revised as requested and are highlighted in red in Figure15 of the manuscript.

Point 3.9: Figure 16 figure not needed, delete it.

Response 3.9: Thanks to the reviewers for your comments, which have been deleted in the manuscript.

Point 3.10: Figure 22 not legible, Authors must correct it.

Response 3.10: Thanks to the reviewers for your comments, which have been revised as requested and are highlighted in red in Figure 21 of the manuscript.

Point 3.11: Line 287 "ferrite and pearlite transform into austenite" - explain the process precisely, it is important because of the topic of work.

Response 3.11: Thanks to the reviewers for your comments, which have been revised as requested and are highlighted in red on line 308 and 370 of the manuscript. The modified sentence is expressed as follows:

During the machining process, high temperature and strain are produced in the cutting area, which causes some ferrite and pearlite transform into austenite.

while the transition area is the area where the metamorphic layer contacts with the sub-strate material, which causes some ferrite and pearlite transform into austenite.

Additional questions Point 1: as a result of high temperature, cracks in the top layer may form, have you observed such damage?

Additional questions Response 1: Thanks to the reviewers for your comments, During the experimental analysis of the gear surface, the presence of adhering particles, pits and surface coating on the surface of the workpiece after cutting was observed, and surface defect behavior can be reduced by means of optimised cutting parameters.

Additional questions Point 2: what was the surface roughness after milling under different conditions?

Additional questions Response 2: Thanks to the reviewers for your comments, a study of the roughness of the machined surface of gears found that, with the increase of cutting speed, the cutter pattern of the machined surface is gradually denser, and the residual height becomes smaller. the machined surface roughness of contour bevel gear gradually increases with the increase of the feed rate.

Reviewer 2 Report

The paper is good and worth reading. However there is some overlapping between another paper which is under review at Chinese journal of mechanical engineering. It is advisable to wait for the paper to be published and then this paper can be presented as extension of the previous work.

Some figures, cutting parameters, conclusions are same. There is change of wordings in both papers.

results related to XRD, EDS and micro hardness testing are similar in both papers.

https://www.researchsquare.com/article/rs-1103600/v1

Additional comments:

  1. The literature review is short. Papers related to FE analysis with same or similar materials should be presented.
  2. Contours of temperature, strain and strain rate at different parameters settings should be presented and compared.
  3. Please explicitly write what is novel in your methodology and conclusions.
  4. Lines 322-324: " This is because......". please elaborate and give scientific reasoning why deformation resistance increase with feed rate.
  5. Milling cutter geometric parameters should be specified along with cutting parameters.
  6. The quality of the machined surface is greatly influenced by tool nose radius and cutting edge preparation. Why these factors are not considered?
  7. Lines 327-333: discussion regarding figure 20, explain the reason for this transition i.e. 0 to 40 and 40 to 90. Give scientific reasoning.
  8. Temperature measurement should be carried out to judge the accuracy of finite element models.

Author Response

Response to Reviewer 2 Comments

Point 0: The paper is good and worth reading. However there is some overlapping between another paper which is under review at Chinese journal of mechanical engineering. It is advisable to wait for the paper to be published and then this paper can be presented as extension of the previous work. Some figures, cutting parameters, conclusions are same. There is change of wordings in both papers. results related to XRD, EDS and micro hardness testing are similar in both papers.

Response 0:

Thanks to the reviewers for your comments, Although the two articles share some equipment usage, they differ in terms of research content, research methods, and research significance.

  1. In terms of research methods, the former mainly investigates the quality of machined surfaces of contour bevel gears under dry milling conditions by observing and analysing the surface morphology, surface defects and machining hardening of the machined surfaces of contour bevel gears. The latter is mainly exploring the cutting parameters on the contour bevel gear milling hardening and metamorphic layer of the machined surface influence through the contour bevel gear single tooth cutting simulation and contour bevel gear cutting machining test.
  2. In terms of research content, the former focuses on the optimisation of tooth quality through the study of the physical characteristics of contour bevel gear tooth surfaces. The latter focuses on the improvement of the physical properties of the metamorphic layer and the machinability of the cutting section by studying the metamorphic layer properties and the machining hardening of the cutting section.
  3. In terms of research significance, the former is mainly carried out by optimising the machined surface quality of contour bevel gears, improving contour bevel gear machining accuracy, reducing friction losses and increasing transmission efficiency; The latter is achieved mainly by improving the physical properties and machinability of the metamorphic layer, achieving an improvement in the smoothness of the gear mesh drive and the fatigue strength of the workpiece.

Point 1: The literature review is short. Papers related to FE analysis with same or similar materials should be presented.

Response 1: Thanks to the reviewers for your comments, which have been revised as requested and are highlighted in red on line 81 and references of the manuscript.

Point 2: Contours of temperature, strain and strain rate at different parameters settings should be presented and compared.

Response 2: Thanks to the reviewers for your comments, which have been revised as requested and are highlighted in red in the Figure 11 and 12 of the manuscript.

Point 3: Please explicitly write what is novel in your methodology and conclusions.

Response 3: Thanks to the reviewers for your comments, which have been revised as requested and are highlighted in red in the conclusion section and line 99 of the manuscript.

Point 4: Lines 322-324: " This is because......". please elaborate and give scientific reasoning why deformation resistance increase with feed rate.

Response 4: From the above finite element simulation, it can be seen that the cutting temperature increases with the increase of feed, part of the ferrite and pearlite into austenite under the effect of cutting heat, so that the surface toughness of the workpiece has increased. Thus the cutting deformation resistance increased in the cutting process, resulting in the degree of hardening of the metamorphic layer machining increased.

Point 5: Milling cutter geometric parameters should be specified along with cutting parameters.

Response 5: Thanks to the reviewers for your comments, which have been revised as requested and are highlighted in red on line 194 of the manuscript. The modified sentence is expressed as follows:

The rake angle of blades is 12°, the tool clearance is 19°34′, the conner radius of the main cutting edge is 1.52mm, the tool point width is 3.45mm and the main pressure angle is 21°19′.

Point 6: The quality of the machined surface is greatly influenced by tool nose radius and cutting edge preparation. Why these factors are not considered?

Response 6: Thanks to the reviewers for your comments, the number of test pieces machined in this experiment is small and the tools used are coated, and this paper focuses on the effect of cutting parameters on the metamorphic layer and work hardening, so the effect of tool nose radius and cutting edge preparation are not considered for the time being.

Point 7: discussion regarding figure 20, explain the reason for this transition i.e. 0 to 40 and 40 to 90. Give scientific reasoning.

Response 7: As can be seen from Figure 20, the machined section is divided into three regions, namely the metamorphic region, the transitional region and the substrate material region. Combined with the work hardening analysis of the metamorphic layer and finite element simulation, it can be found that the change in material temperature, stress and hardness is large when moving from the metamorphic layer region to the transition region, and relatively small when moving from the transition region to the substrate material region. Therefore, from Figure 19, it can be found that for 0-40μm, the machined section is located in the metamorphic region and in the transition region, where the slope of the curve is larger, while for 40-90μm, the machined section is located in the transitional region and in the substrate material region, where the slope of the curve decreases, thus dividing Figure 19 into two regions, 0-40μm and 40-90μm.

Point 8: Temperature measurement should be carried out to judge the accuracy of finite element models.

Response 8: Thanks to the reviewer's comments, the temperature experiments could not be performed due to the COVID-19, but the manuscript was compared with the experiments through Figure 19 to verify the accuracy of the simulations, which will be considered subsequently when available.

Round 2

Reviewer 1 Report

Thanks to the authors for improving most of my comments. However, the quality of all the drawings is still insufficient. The  all drawings are illegible, with many blurring and Chinese characters.

Author Response

Response to Reviewer 1 Comments

Point 1: Thanks to the authors for improving most of my comments. However, the quality of all the drawings is still insufficient. The  all drawings are illegible, with many blurring and Chinese characters.

Response 1: Thanks to the reviewers for your comments, which have been revised as requested and are highlighted in red in the appropriate places in the manuscript. Thanks again to the reviewers for your comments and suggestions.

Reviewer 2 Report

Most of the changes suggested are covered by authors. The paper can be accepted now.

Author Response

Response to Reviewer 2 Comments

Point 1: Most of the changes suggested are covered by authors. The paper can be accepted now.

Response 1:

Thanks to the reviewers for your comments, we are delighted to hear this and wish you well in your work.
